# Structures, Electronic, and Magnetic Properties of CoK*_n_* (*n* = 2–12) Clusters: A Particle Swarm Optimization Prediction Jointed with First-Principles Investigation

**DOI:** 10.3390/nano13152155

**Published:** 2023-07-25

**Authors:** Yi Jiang, Maidina Aireti, Xudong Leng, Xu Ji, Jing Liu, Xiuhua Cui, Haiming Duan, Qun Jing, Haibin Cao

**Affiliations:** 1Xinjiang Key Laboratory of Solid State Physics and Devices, Xinjiang University, 777 Huarui Road, Urumqi 830017, China; jiangyi@xju.edu.cn (Y.J.); madena1227@163.com (M.A.); 107552100701@xju.edu.cn (X.L.); jx@xju.edu.cn (X.J.); xdlj@xju.edu.cn (J.L.); qunjing@xju.edu.cn (Q.J.); 2School of Physical Science and Technology, Xinjiang University, 777 Huarui Road, Urumqi 830017, China; 3Department of Physics, College of Sciences, Shihezi University, Shihezi 832000, China; caohb@shzu.edu.cn

**Keywords:** CoK*_n_* (*n* = 2–12) clsuters, first-principles investigation, magnetic moments, electronic structures

## Abstract

Transition-metal-doped clusters have long been attracting great attention due to their unique geometries and interesting physical and/or chemical properties. In this paper, the geometries of the lowest- and lower-energy CoK*_n_* (*n* = 2–12) clusters have been screened out using particle swarm optimization and first principles relaxation. The results show that except for CoK_2_ the other CoK*_n_* (*n* = 3–12) clusters are all three-dimensional structures, and CoK_7_ is the transition structure from which the lowest energy structures are cobalt atom-centered cage-like structures. The stability, the electronic structures, and the magnetic properties of CoK*_n_* clusters (*n* = 2–12) clusters are further investigated using the first principles method. The results show that the medium-sized clusters whose geometries are cage-like structures are more stable than smaller-sized clusters. The electronic configuration of CoK*_n_* clusters could be described as 1S1P1D according to the spherical jellium model. The main components of petal-shaped D molecular orbitals are Co-d and K-s states or Co-d and Co-s states, and the main components of sphere-like S molecular orbitals or spindle-like P molecular orbitals are K-s states or Co-s states. Co atoms give the main contribution to the total magnetic moments, and K atoms can either enhance or attenuate the total magnetic moments. CoK*_n_* (*n* = 5–8) clusters have relatively large magnetic moments, which has a relation to the strong Co-K bond and the large amount of charge transfer. CoK_4_ could be a magnetic superatom with a large magnetic moment of 5 μ_B._

## 1. Introduction

During the past decades, clusters have drawn significant attention due to their unique geometries, interesting physical and/or chemical properties originating from quantum size effects, and their wide application in catalysts [1], superatoms [2], and so on [3,4,5,6]. For example, Cheng et al. fabricated single platinum atoms and clusters using the atomic layer deposition (ALD) technique and then investigated the activity of hydrogen evolution reaction (HER). The results show that the platinum single atoms and clusters supported on nitrogen-doped graphene nanosheets have greater activity for HER due to the high utilization of nearly all the platinum atoms [1]. Hence, a large number of investigations have been performed to explore the geometries, stabilities, electronic structures, and physical properties of clusters [7,8,9,10] using such methods as the first principles investigation [11,12,13], molecular dynamics [14,15], bionics algorithm [16,17,18], and machine learning [19,20,21], among which kinds of bionics algorithm are on the rise.

The structures and physical properties of transition-metal-doped clusters including the transition-metal oxides and hydroxides nanoclusters are one of the hotspots [22,23,24,25]. Wang et al. investigated the total and local magnetic moments of the most stable [TM_13_@Au_20_]^−^ clusters using density functional theory implemented in the DMol^3^ package, and they pointed out that the transition-metal atom could enhance or attenuate the total magnetic moments [26]. Zhao et al. investigated the geometries and electronic structures of Y*_n_*Al (*n* = 1–14) clusters and found that doping with an Al atom could attenuate the magnetic moments but enhance the stabilities of the yttrium framework [27]. Our previous work predicted the geometries of the lowest energy and their isomers of ScK*_n_* (*n* = 2–12) clusters using particle swarm optimization and first principles geometry optimization, and we have also investigated the electronic structures and magnetic properties using the first principles calculation. The results showed that the doping of Sc atoms can improve the magnetic properties and stability of the K*_n_* clusters, and the ScK_12_ cluster may be a magnetic superatom with a high magnetic moment of 5 μ_B_ [28]. The VIIIB atom (Fe, Co, and Ni)-doped clusters including VIIIB transition-metal oxides and hydroxides nanoclusters have also attracted great attention due to their interesting electronic structures, physical and chemical properties, and their wide application. For example, Milan Babu Poudel et al. reported that the hierarchical heterostructure comprising ternary metal sulfides covered by nickel–cobalt layered double hydroxide could be used as binder-free cathode material for supercapacitor application [29], and they also reported a superior bifunctional electrocatalyst for oxygen evolution reaction (OER) and hydrogen evolution reaction (HER), which contains both cobalt and nickel atoms [30]. As for the VIIIB atom (Fe, Co, and Ni)-doped clusters, our group has investigated the geometries, electronic structures, and magnetic properties of Ge*_n_*Co (*n* = 2–12) clusters using the first principles method, and the results show that the total magnetic moments of Ge*_n_*Co (*n* = 2–12) clusters does not quench, and the doping of the Co atom is beneficial to enhance the stability of host Ge_n_ clusters [31]. The electronic structures and magnetic moments of the core-shell clusters Co_13_@TM_20_ (TM = Mn, Fe, Co, and Ni) have also been investigated using the first-principles method, and the results show that these clusters have huge magnetic moments, especially for the Co_13_@Mn_20_ cluster, whose magnetic moment is as large as 113 μ_B_ [32]. After investigating the geometries and electronic structures using the first-principles method implemented in the Vienna Ab Initio Simulation Package (VASP) and Orca code, Hao et al. pointed out that doping of the Co atom in B*_n_* clusters significantly changes their structures, and the Co_2_B and Co_2_B_7_ clusters have a large magnetic moment of 3 μ_B_ [33]. A systematic theoretical investigation of the structure, stability, and electronic properties of Li*_n_*Co clusters showed that the doping with the cobalt atom could enhance the stabilities of host clusters, and greater electron transfer from Li-2s to Co-3d can help to strengthen the bond length of Li-Co [34].

Inspired by the unique structures, interesting electronic properties, and wide application of transition-metal-doped clusters, the authors are going to investigate the growth pattern, electronic structures, and magnetic moments of other VIIIB atom-doped alkali clusters. In this paper, the authors have investigated the geometries of CoK*_n_* (*n* = 2–12) clusters predicted by the particle swarm optimization (PSO) algorithm along with first principles relaxation. The results show that the CoK*_n_* (*n* = 3–12) clusters are all three-dimensional structures, and CoK_7_ is the transition structure from which the lowest energy structures are cobalt atom-centered cage-like structures. The stabilities, the magnetic moments, and the electronic structures are also investigated using the first principles calculation. The results show that the cobalt atom-centered cage-like clusters are more stable than smaller-sized clusters. In addition, the electronic configuration of CoK*_n_* clusters could be described as 1S1P1D according to the spherical jellium model. According to the results of the projected density of states, the main components of petal-shaped D molecular orbitals are Co-d and K-s states or Co-d and Co-s states. The main components of sphere-like S molecular orbitals or spindle-like P molecular orbitals are K-s states or Co-s states. Co atoms give the main contribution to the total magnetic moments, and K atoms can either enhance or attenuate the total magnetic moments. CoK*_n_* (*n* = 5–8) clusters have relatively large magnetic moments, which has a relation with the strong Co-K bond and a large amount of charge transfer. CoK_4_ could be a magnetic superatom with a large magnetic moment of 5 μ_B._

## 2. Computational Details

In this paper, the particle swarm optimization method implemented in CALYPSO code [16,35,36] along with the first principles calculation implemented in the Vienna Ab Initio Simulation Package (VASP) code [37,38] are used to screen out the potential candidates of CoK_n_ (*n* = 2–12) clusters. The CALYPSO code has been widely used to predict the ground state structure of clusters and crystals [39,40,41]. Using this powerful structure search method combined with first-principles calculations, Tang et al. have identified a planar CoB_6_ monolayer as a stable two-dimensional ferromagnet [42], and our group has also investigated the geometries of transition-metal-doped clusters [43].

During the workflow of cluster structure prediction, the candidate clusters of the first generation are randomly generated by CALYPSO code, and then 80% of candidate clusters of the next generation are inherited from the previous generation. The remaining 20% of candidate clusters are still randomly generated. The population size, i.e., the total number of structures per generation, is set to be 20 for CoK*_n_* (*n* < 9) and 30 for CoK*_n_* (*n* ≥ 9) clusters. The maximum number of generations is set as more than 20 (for *n* < 9) and 30 (for *n* ≥ 9), that is to say, more than 400 (for *n* < 9) and 900 (for *n* ≥ 9) candidate isomers are generated and optimized to get the lowest-energy clusters. The geometry optimization is performed using the VASP code [35,36]. During the calculation, the projected augmented wave method and the Perdew–Burke–Ernzerhof (PBE) exchange-correlation functional [44] under generalized gradient approximation (GGA) are used. The plane-wave cutoff energy is set as 300 eV. The global break condition for the electronic self-consistency (SC) loop is set as 1 × 10^−5^ eV. The maximum number of electronic SC steps is set as 500, and the maximum number of ionic steps is set as 200. A conjugate gradient algorithm is used to relax the ions into their instantaneous ground state. To avoid the interaction of atoms located in neighbor cells, the CoK*_n_* clusters are placed in a cubic super lattice of 12 × 12 × 12 angstroms (for *n* < 9) and 15 × 15 × 15 angstroms (for *n* ≥ 9).

Once the convergence criterion is obtained, the PSO performance will be stopped, and then some candidates are chosen to be further optimized using the first principles Gaussian09 code. During Gaussian09 calculation [45], the geometry optimization and frequency calculation of CoK*_n_* (*n* = 2–12) clusters with different spin are carried out to ensure that the obtained structures are the lowest-energy structures. The Gaussian09 calculation is performed using B3LYP [46] functional at a 6-31G** level for the Co atom and a 6-311G** [47] level for K atoms. The authors have carefully checked out the harmonic vibrational frequencies of each lowest- and lower-energy clusters to ensure the obtained clusters are stable clusters without imaginary frequencies. The population analysis of CoK_n_ clusters is studied using the Multiwfn3.8 program [48,49].

## 3. Results and Discussion

### 3.1. The Geometries of Lowest-Energy Clusters

Using the method described above, the geometries of the lowest-energy clusters and their meta stable isomers are obtained. Figure 1 gives the geometries of the lowest-energy and the meta stable isomers of CoK*_n_* (*n* = 2–12) clusters along with the bond length of Co-K bonds. As shown in Figure 1, except for CoK_2_ clusters whose geometries of lowest-energy and their isomer are planar triangular structures, the geometries of the other CoK*_n_* (*n* = 3–12) clusters are all three-dimensional structures. The geometries of the ground state and meta stable state of CoK_3_ cluster are a slightly distorted quadrangle with C_2V_ symmetry. The bipyramid structures were found in the lowest energy structures of CoK*_n_* (*n* = 4–6) clusters. The lowest energy structure of CoK_4_ is a triangular bipyramid structure with a C_3V_ point group. The lowest energy structure of cluster CoK_5_ is a tetragonal bipyramid structure with a C_4V_ point group. The CoK_6_ is a twisted pentagonal bipyramid structure with a C_1_ point group. The ground state structure of the CoK_7_ cluster is a Co-atom-centered K-atom-capped tetragonal bipyramid structure with a C_3V_ point group. Noting that from the CoK_7_ cluster, the lowest-energy structure becomes a Co-atom-centered polyhedron. The lowest-energy structure of the CoK_8_ cluster is a Co-atom-centered K-capped pentagonal bipyramid with a C_s_ point group. At first glance, the lowest-energy structure of the CoK_9_ cluster looks like a Co-atom-centered K-atom-capped square antiprism, but actually, it is a Co-atom-centered three-K-atom-capped trigonal prism because its point group is D_3h_. The lowest-energy structure of the CoK_10_ cluster is a Co-atom-centered two-K-atom-capped square antiprism. It can also be described as a Co-atom-centered 1-4-4-1-layered structure. The lowest-energy structure of the CoK_11_ cluster is a distorted Co-atom-centered cage-like structure. The lowest energy structure of the CoK_12_ cluster is a distorted Co-atom-centered icosahedral structure with a C_1_ point group.

In a word, along with the increased size of clusters, the lowest-energy structures transform from planar geometry to dense packing structures, and the Co atom moves from the apex position (*n* < 7) to the central position of cage-like structures (*n* ≥ 7). A similar growth pattern can also be found in other VIIIB atom (Fe, Co, and Ni)-doped clusters. For Ge*_n_*Co (*n* = 1–13) clusters, the geometries of the most stable Ge*_n_*Co (*n* = 1–3) clusters are planar geometries. The geometries of most stable medium-sized Ge*_n_*Co (*n* = 4–13) clusters are three-dimensional configurations, and from the Ge_9_Co cluster, the dopant cobalt atom encapsulated geometry becomes the lowest-energy structure. As for Li*_n_*Co (*n* = 1–12) clusters, the smallest Li*_n_*Co (*n* = 1–3) clusters also adopt planar geometries, and the medium-sized Li*_n_*Co (*n* = 4–12) clusters adopt three-dimensional configurations. Like the results shown in this paper, from the Li_7_Co cluster, the most stable Li*_n_*Co clusters adopt cobalt atom-centered cage-like structures.

### 3.2. Relative Stability

The stability of CoK*_n_* (*n* = 2–12) clusters is further evaluated by the binding energy per atom (*E_b_*), the second-order difference of energies (Δ_2_*E*), and fragmentation energies (*E_f_*), which are defined as follows:(1)$$E_{b}=\frac{\left[E\left(Co\right)+nE\left(K\right)-E\left({CoK}_{n}\right)\right]}{\left(n+1\right)}$$
(2)$${∆}_{2}E=E\left({CoK}_{n+1}\right)+E\left({CoK}_{n-1}\right)-2E\left({CoK}_{n}\right)$$
(3)$$E_{f}=E\left({CoK}_{n-1}\right)+E\left(K\right)-E\left({CoK}_{n}\right)$$
where *E* represents the total energy of the cluster or atoms marked in parentheses.

As shown in Figure 2, the binding energy per atom gradually increases until the CoK_9_ cluster has the largest binding energy per atom, and then size dependence becomes smooth for *n* = 9~12. The results show that the large-sized clusters are more stable than small clusters.

As for the second-order difference of energies (Δ_2_*E*) and fragmentation energies (*E_f_*), they own similar curves, as shown in Figure 2. The local peaks of Δ_2_*E* localized at *n* = 4, 6, 9, and the local peaks of *E_f_* are found at *n* = 4, 6, 8, 11, implying these clusters are more stable than their neighbors.

According to the obtained results shown in Figure 2, the authors believe the CoK_4_, CoK_6_, CoK_8_, and CoK_9_ clusters are more stable than their neighbors. The relatively strong stability may have a relation with the relatively smaller bond length and relatively strong atomic interactions. For example, the bond lengths of Co-K in CoK_4_ are 2.97, 2.97, 2.97, and 3.73 angstrom, respectively, which are smaller than the bond length of Co-K in CoK_5_ (about 3.20, 3.20, 3.34, 3.34, and 4.60 angstrom). The bond lengths of Co-K in CoK_6_ are 2.94, 2.97, 3.00, 4.25, and 5.41 angstrom, which are also smaller than their neighbor clusters. Noting that although the cage-like clusters have relatively large bond lengths, the dense packing cage-like structures make these clusters have a relatively large binding energy per atom. Similar conclusions can also be found in other VIIIB atom-doped clusters.

### 3.3. Magnetic Properties

It is interesting to investigate the magnetic properties of transition-metal atom-doped clusters because the doping of a transition-metal atom can enhance or attenuate total magnetic moments [26,31,32], and the stable cluster can also be used as a superatom with giant or enhanced magnetic moments [28,50]. Herein, the total magnetic moments of CoK*_n_* (*n* = 2–12) clusters are obtained using the Gaussian09 calculation at the B3LYP/6-31G**(Co) and B3LYP/6-311G**(K) levels. The obtained total magnetic moments of CoK*_n_* (*n* = 2–12) clusters are shown in Figure 3 and Table 1. As shown in Figure 3 and Table 1, the magnetic moment of CoK*_n_* (*n* = 2–4) rapidly increases from 1 μ_B_ to 5 μ_B_, and then gradually decreases to 1 μ_B_, except for CoK_7_ and CoK_11_, whose magnetic moments are 4 μ_B_ and 2 μ_B_, respectively. The magnetic moments of CoK*_n_* (*n* = 4–12) are 5 μ_B_ (CoK_4_), 4 μ_B_ (CoK_5_), 3 μ_B_ (CoK_6_), 4 μ_B_ (CoK_7_), 3 μ_B_ (CoK_8_), 2 μ_B_ (CoK_9_), 1 μ_B_ (CoK_10_), 2 μ_B_ (CoK_11_), and 1 μ_B_ (CoK_12_), respectively.

To deeply understand the atomic contribution to the total magnetic moments, the local magnetic moments of cobalt and potassium atoms are further analyzed by the natural electronic configuration (NEC) and Mulliken population analysis. The obtained local magnetic moments of cobalt and potassium atoms are listed in Figure 3 and Table 1. As shown in Figure 3 and Table 1, for all CoK*_n_* (*n* = 2–12) clusters, the atomic magnetic moment of the Co atom is in the range of 2.1937 μ_B_ to 2.9430 μ_B_, indicating that the Co atom plays an important role in determining the total magnetic moment. As for K atoms, they can enhance (as in the case of CoK_4_, CoK_5_, CoK_6_, CoK_7_, and CoK_8_) or attenuate (as in the case of CoK_2_, CoK_3_, CoK_9_, CoK_10_, CoK_11_, and CoK_12_) the total magnetic moments of CoK*_n_* clusters.

As described above, the CoK_4_ cluster exhibits the largest magnetic moment among all these clusters, hence the authors would investigate the molecular orbitals to dig out the origination of the largest magnetic moment. The obtained molecular orbitals are shown in Figure 4. For comparison, the molecular orbitals of CoK_5_ are also investigated. As shown in Figure 4, the molecular orbitals look like sphere-like *s* orbitals, spindle-like *p* orbitals, and petal-shaped *d* orbitals. For those petal-shaped orbitals, the electrons localize around the Co atom, and the sphere-like and spindle-like electrons delocalize around all atoms. According to the data shown in Figure 4, the authors believe their electrons can be described as 1S^2^1P^3^1D^8^ (for CoK_4_) and 1S^2^1P^4^1D^8^ (for CoK_5_) according to the spherical jellium model [45,46].

The projected density of states (PDOS) is also utilized to investigate the atomic interaction and atomic contribution to the molecular orbitals of CoK*_n_* clusters. Figure 5 gives the spin-polarized projected density of states of CoK_4_ and CoK_5_ clusters. Take CoK_4_, for example. As shown in Figure 4, the molecular orbitals of CoK_4_ are found in the energy range of −2.0~−2.8 eV (spindle-like spin-up molecular orbitals), −3.4~−4.6 eV (petal-shaped spin-up molecular orbitals), −4.8 eV (sphere-like spin up molecular orbital), −2.0~−2.4 eV (petal-shaped spin-down molecular orbitals), and −4.0 eV (sphere-like spin-down molecular orbitals). As shown in Figure 5, in the energy range of −2.0~−2.8 eV, there are K-s, Co-s, and Co-p states (for spin-up states), and K-s, Co-d states (for spin-down states). That is to say, the hybrid molecular orbitals coming from the Co-K interaction are found in this energy range, and the K-s states play an important role in forming the spindle-like spin-up molecular orbitals. The Co-d and K-s atomic states give the main contribution to forming the spin-down petal-shaped molecular orbitals. In the energy range of −3.0~−4.0 eV, there are mainly the Co-d and K-s states in the spin-up states, indicating the spin-up petal-shaped molecular orbitals are mainly coming from the Co-d and K-s atomic states. While there are only Co-s states found nearby −4 eV in spin-down molecular orbitals, indicating the Co-s states give the main contribution to determining the spin-down molecular orbitals. There are Co-d and Co-s states found in the energy range of −4.0~−4.6 eV in the spin-up molecular orbitals, and there are mainly Co-s states found in the spin-up molecular orbitals nearby −4.8 eV. So, the Co-d and Co-s states give the main contribution in forming the spin-up petal-shaped molecular orbitals in the energy range of −4.0~−4.6 eV, and Co-s states determine the spin-up spherical molecular orbitals nearby −4.8 eV. In a word, as for the CoK_4_ cluster, the main components of petal-shaped D molecular orbitals come from Co-d and K-s states (such as D molecular orbitals in the energy range of −2.0~−2.4 eV) or Co-d and Co-s states (such as D molecular orbitals in the energy range of −4.0~−4.6 eV). In addition, the main components of sphere-like S molecular orbitals or spindle-like P molecular orbitals are K-s states (such as P molecular orbitals in the energy range of −2.0~−2.8 eV) or Co-s states (such as S molecular orbitals nearby −4.8 eV). Similar conclusions can also be found in the CoK_5_ cluster. The main components of spindle-like P molecular orbitals in the energy range of −2.0~−2.8 eV are K-s states. The main components of spherical S molecular orbitals nearby −4.3 eV and −4.7 eV are Co-s states. The main components of petal-shaped D molecular orbitals in the energy range of −3.6~−5.0 eV are Co-d and K-s states (for D orbitals in the energy range of −3.6~−4.2 eV) and Co-d and Co-s states (for D molecular orbitals in the energy range of −4.6~−5.0 eV).

The NEC and Mulliken atomic charges show that the Co atom has negative atomic charges ranging from −0.2175 e to −0.9731 e, indicating the Co atom is a charge acceptor, and K atoms are charge donors. Although both K and Co atoms are metal elements, the latter has larger Pauling’s electronegativity (1.88 for the Co atom, and 0.82 for the K atom) which makes the Co atom accept electrons transferred from K atoms. Note that the CoK_4_ cluster has the shortest average bond length of the Co-K bond and a relatively large atomic charge of the Co atom (about −0.8717 e). The electronic configuration of the isolated Co atom is [Ar]3d^7^4s^2^. In the CoK*_n_* clusters, the *spd* hybridization was found (as discussed above) and the hybridization makes the electron transfer from the Co-4s states and K-2s states to the Co-3d states and Co-4p states (shown in the NEC results listed in Table 1, and the spin-polarized projected density of states shown in Figure 5). The strong Co-K interaction and a large amount of charge transfer result in the enhanced magnetic moment of the CoK_4_ cluster. Similar to the CoK_4_ cluster, enhanced magnetic moments are also found in CoK*_n_* (*n* = 5–8) clusters. The relatively weak interaction of Co-K coming from the relatively large bond length of Co-K bond in other CoK*_n_* clusters (especially the medium-sized cage-like clusters) makes these clusters have relatively smaller magnetic moments.

## 4. Conclusions

In this paper, the geometries of CoK*_n_* (*n* = 2–12) clusters are predicted using the PSO method joined with the first-principles geometries optimization, and then the electronic structures and magnetic moments are further investigated using the DFT calculation. The results show that the lowest-energy structures transform from planar geometry to dense packing structures, and the Co atom moves from the apex position (*n* < 7) to the central position of cage-like structures (*n* ≥ 7). Medium-sized clusters with cage-like geometries are more stable than small clusters, and CoK_4_, CoK_6_, CoK_8_, and CoK_9_ clusters are more stable than their neighbors. The electronic configuration of CoK*_n_* clusters can be described as 1S1P1D according to the spherical jellium model. The main components of petal-shaped D molecular orbitals are Co-d and K-s states or Co-d and Co-s states. The main components of sphere-like S molecular orbitals or spindle-like P molecular orbitals are K-s states or Co-s states. The Co atom plays an important role in determining the total magnetic moments, and K atoms can either enhance or attenuate the total magnetic moments. CoK*_n_* (*n* = 5–8) clusters have relatively large magnetic moments which originate from the strong interaction of Co-K and a large amount of charge transferring from K to Co atoms. CoK_4_ could be a magnetic superatom with a large magnetic moment of 5 μ_B._

## Figures and Tables

**Figure 1 nanomaterials-13-02155-f001:**
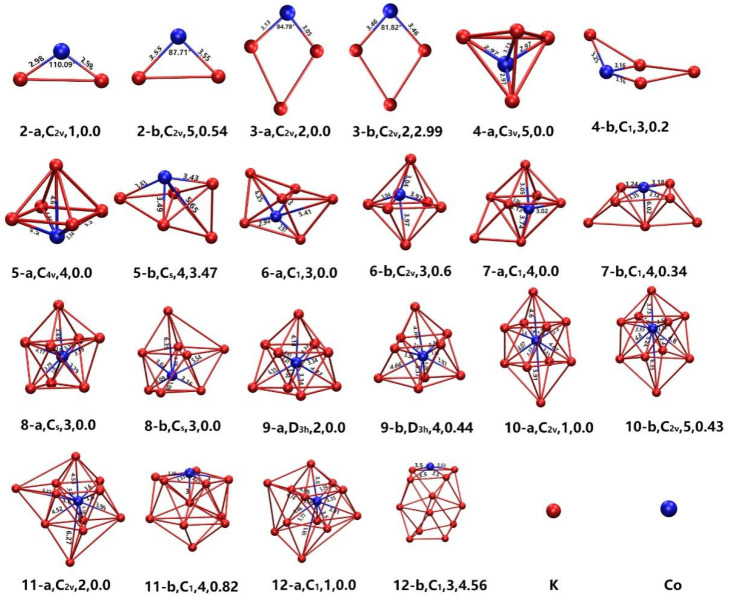
The ground state structure of CoK*_n_* (*n* = 2–12) clusters (marked as n-a, such as 12-a) and their meta-stable isomers (marked as n-b, such as 12-b). The text below the image also gives the point group, magnetic moment (in μ_B_), and the relative energy (in eV) compared with the ground state.

**Figure 2 nanomaterials-13-02155-f002:**
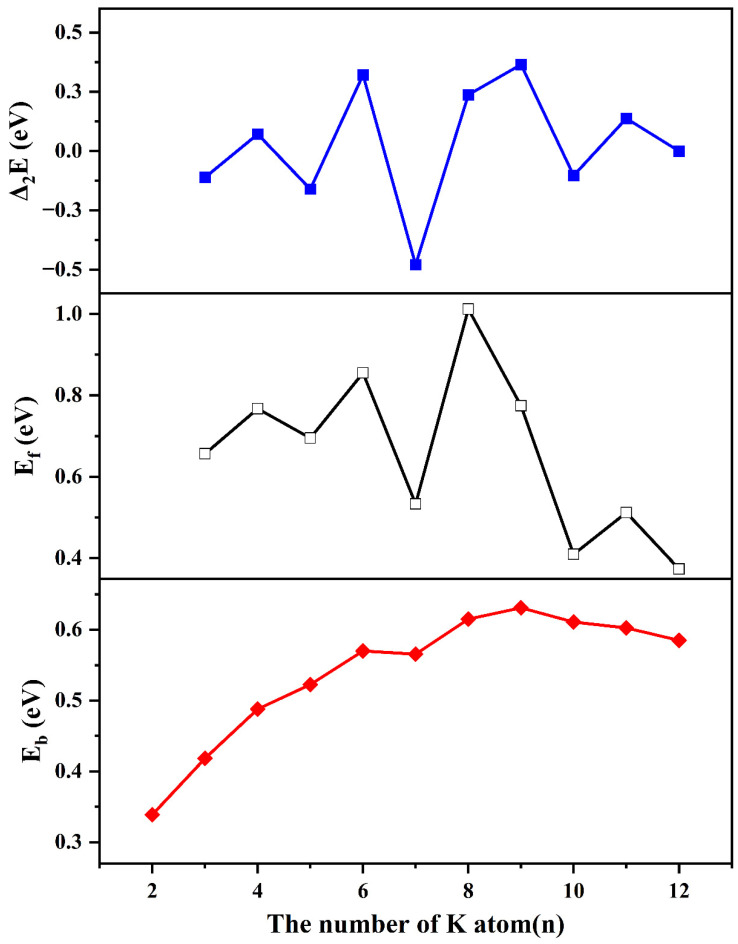
The binding energy per atom (*E_b_*), the second-order difference of energies (Δ_2_*E*), and fragmentation energies (*E_f_*) of CoK*_n_* (*n* = 2–12) clusters.

**Figure 3 nanomaterials-13-02155-f003:**
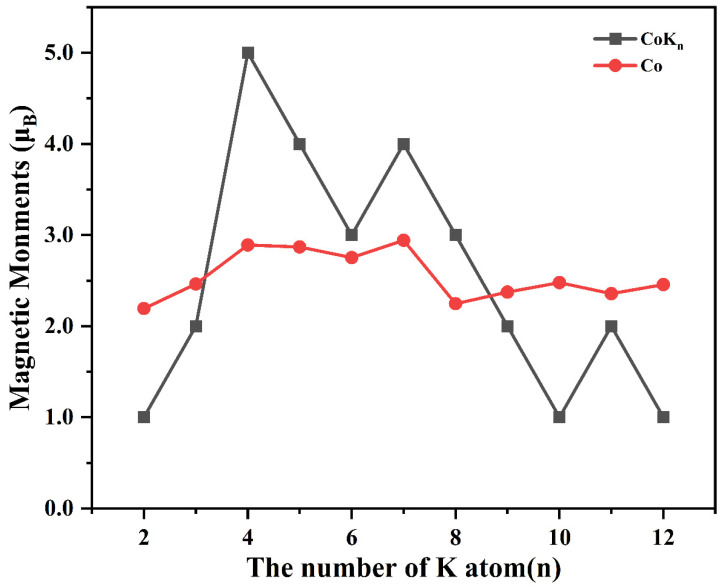
The total magnetic moments of CoK*_n_* (*n* = 2–12) clusters (black) and local magnetic moments of Co atom in CoK*_n_* (*n* = 2–12) clusters (red).

**Figure 4 nanomaterials-13-02155-f004:**
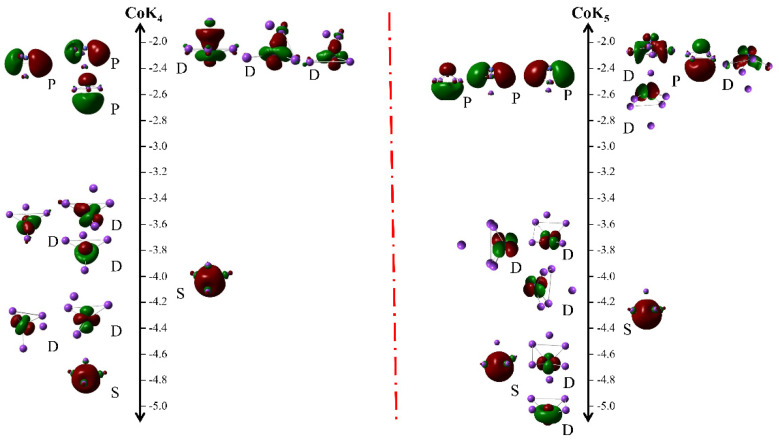
The molecular orbitals of CoK_4_ (**left**) and CoK_5_ (**right**) clusters.

**Figure 5 nanomaterials-13-02155-f005:**
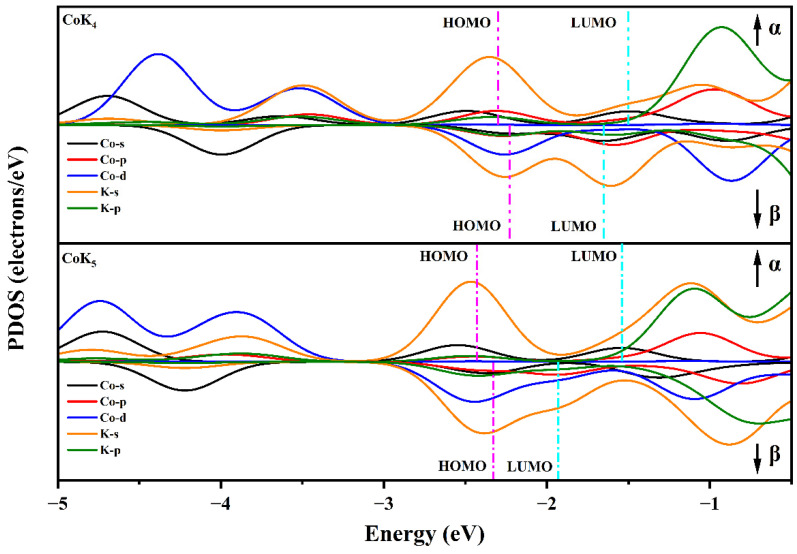
The projected density of states (PDOS) of CoK_4_ (**upper**) and CoK_5_ (**bottom**) clusters.

**Table 1 nanomaterials-13-02155-t001:** The natural electron configuration of Co atom (NEC(Co)), the magnetic moments of CoK*_n_* clusters (m(CoK*_n_*)), Co atom (m(Co)), K atoms (m(K)), the Mulliken atomic charges of Co (q(Co)), and the average bond length of Co-K bond (R(Co-K)).

Cluster	NEC(Co)	m(CoK*_n_*)	m(Co)	m(K)	q(Co)	R(Co-K)
CoK_2_	4s^1.74^3d^7.45^4p^0.31^	1	2.1937	−1.1937	−0.4264	2.98
CoK_3_	4s^1.83^3d^7.43^4p^0.18^	2	2.4643	−0.5186	−0.2175	4.08
CoK_4_	4s^1.68^3d^7.51^4p^1.16^	5	2.8923	2.1077	−0.8717	2.91
CoK_5_	4s^1.8^3d^7.4^4p^1.06^	4	2.8704	1.1418	−0.2659	3.54
CoK_6_	4s^1.73^3d^7.56^4p^1.2^	3	2.7535	0.2465	−0.8875	3.69
CoK_7_	4s^1.71^3d^7.48^4p^2.02^	4	2.9430	1.0570	−0.9731	3.63
CoK_8_	4s^1.67^3d^7.43^4p^2.64^	3	2.2473	0.7501	−0.7206	3.53
CoK_9_	4s^1.91^3d^7.39^4p^2.68^	2	2.3750	−0.3750	−0.4794	3.65
CoK_10_	4s^1.84^3d^7.4^4p^2.53^	1	2.4785	−1.4785	−0.4925	3.89
CoK_11_	4s^1.79^3d^7.42^4p^2.44^	2	2.3572	−0.3572	−0.4844	4.16
CoK_12_	4s^1.78^3d^7.41^4p^2.46^	1	2.4580	−1.4673	−0.5966	4.26

## Data Availability

The datasets generated and/or analysed during the current study are available from the corresponding author on reasonable request.

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
