# Peer review of "Structures, Electronic, and Magnetic Properties of CoKn (n = 2–12) Clusters: A Particle Swarm Optimization Prediction Jointed with First-Principles Investigation"

_nanomaterials, 2023, doi:10.3390/nano13152155_

Round 1

Reviewer 1 Report

The paper describing the stability,  the electronic and magnetic properties of the CoKn clusters (n=2-12) is well written, with a comprehensive introduction, clear description of the theoretical methods and well structured presentation of the results and conclusions. The research is oriented to find the most stable clusters as well as their magnetic moment behaviour and electronic configuration vs. nr. of K atoms in the cluster. 

The following issues are raised:

-In Fig. 3 , by comparing the magnetic moments of CoKn and Co clusters is not clear where from are the Co clusters magnetic moments (calculated from this study, other study or experimental). Please add more information about this in the reviewed version.  Also, please correct the legends of the axis in Fig. 3. 

-The comparison between the obtained results and the results of other investigations (either related to the most stable clusters or the values of their magnetic moments) in the section "Results and discussions" would add much to the value of the paper. Only very few links to the references (similar previous investigations) are present at the beginning of sect. 3.3. 

The reviewer is recommending the publication of the present paper with minor corrections. Congratulations for your work.  

Author Response

The paper describing the stability, the electronic and magnetic properties of the CoKn clusters (n=2-12) is well written, with a comprehensive introduction, clear description of the theoretical methods and well structured presentation of the results and conclusions. The research is oriented to find the most stable clusters as well as their magnetic moment behaviour and electronic configuration vs. nr. of K atoms in the cluster.

The following issues are raised:

-In Fig. 3, by comparing the magnetic moments of CoKn and Co clusters is not clear where from are the Co clusters magnetic moments (calculated from this study, other study or experimental). Please add more information about this in the reviewed version.  Also, please correct the legends of the axis in Fig. 3.

Reply: Thanks very much.

The total magnetic moments of CoKn clusters are obtained using the first principles Gaussian09 calculation within B3LYP functional and 6-31G**(for Co atom) and 6-311G**(for K atoms) basis sets. And the local magnetic moments of Co and K atoms are obtained using natural electronic configuration (NEC) and Mulliken population analysis.

In the revised manuscript, the authors give more clear descriptions as ‘Herein the total magnetic moments of CoKn (n = 2 -12) clusters are obtained using the Gaussian09 calculation at B3LYP/6-31G**(Co) and B3LYP/6-311G**(K) level. The obtained total magnetic moments of CoKn ( n = 2 -12 ) clusters are shown in Figure 3 and Table 1’ and ‘To deeply understanding the atomic contribution to the total magnetic moments, the local magnetic moments of cobalt and potassium atoms are further analyzed by the natural electronic configuration (NEC) and Mulliken population analysis. The obtained local magnetic moments of cobalt and potassium atoms are listed in Figure 3 and Table 1.

-The comparison between the obtained results and the results of other investigations (either related to the most stable clusters or the values of their magnetic moments) in the section "Results and discussions" would add much to the value of the paper. Only very few links to the references (similar previous investigations) are present at the beginning of sect. 3.3.

Reply: Thanks very much.

In the revised manuscript, the authors give more descriptions to compare the results obtained in other papers and this manuscript. For example, as shown in section 3.1, ‘Similar growth pattern can also be found in other VIIIB atom (Fe, Co, and Ni) doped clusters. For GenCo (n = 1 – 13) clusters, the geometries of the most stable GenCo (n = 1 – 3) clusters are planar geometries, the geometries of most stable medium sized GenCo (n = 4 – 13) clusters are three dimensional configurations, and from Ge9Co cluster the dopant cobalt atom encapsulated geometry become the lowest energy structure. As for LinCo (n = 1 – 12) clusters, the smallest LinCo ( n = 1 -3 ) clusters also adopt planar geometries, and the medium sized LinCo (n = 4 – 12) clusters adopt three dimensional configurations. Like the results shown in this paper, from the Li7Co cluster the most stable LinCo clusters adopt cobalt atom centered cage-like structures.

The reviewer is recommending the publication of the present paper with minor corrections. Congratulations for your work. 

Reply: Thanks very much.

Reviewer 2 Report

This manuscripts reports on the Structures, Electronic and Magnetic Properties of CoKn (n=2-12)  Clusters: A particle swarm optimization prediction jointed with  first-principle investigation. Some findings are interesting. However, following minor revisions are suggested:

1.       English language should be polished. Avoid continuous references. Abstract should be concise highlighting the main objectives.

2.       Rational design of cobalt oxides and hydroxide also should be explained with the reference of following articles: Journal of Alloys and Compounds, Volume 960, 15 October 2023, 170678,

Journal of Energy Storage, Volume 60, April 2023, 106713.

3.       DOS profile should be presented.

English language should be polished. 

Author Response

This manuscripts reports on the Structures, Electronic and Magnetic Properties of CoKn (n=2-12) Clusters: A particle swarm optimization prediction jointed with first-principle investigation. Some findings are interesting. However, following minor revisions are suggested:

1.English language should be polished. Avoid continuous references. Abstract should be concise highlighting the main objectives.

Reply: Thanks very much.

According to the reviewer’s suggestion, the authors have polished the manuscript (marked in red in the revised manuscript).

2.Rational design of cobalt oxides and hydroxide also should be explained with the reference of following articles: Journal of Alloys and Compounds, Volume 960, 15 October 2023, 170678,

Journal of Energy Storage, Volume 60, April 2023, 106713.

Reply: Thanks very much.

In the revised manuscript, the authors cite these two articles, and give some comments as ‘The VIIIB atom (Fe, Co, and Ni) doped clusters including VIIIB transition-metal oxides and hydroxides nanoclusters have also attracted great attention due to their interesting electronic structures, magnetic properties and their application. For example, Milan Babu Poudel et al. reported that the hierarchical heterostructure comprising ternary metal sulfides covered by nickel cobalt layered double hydroxide could be used as binder free cathode material for supercapacitor application49, and they also reported a superior bifunctional electrocatalyst for oxygen evolution reaction (OER) and hydrogen evolution reaction (HER) which containing both cobalt and nickel atoms50’.

These two articles are listed as Ref.49 and Ref.50.

  1. Milan Babu Poudel, Allison A. Kim, Prakash Chandra Lohani, Dong Jin Yoo, and Han Joo Kim, Assembling zinc cobalt hydroxide/ternary sulfides hetereostructure and iron oxide nanorods on three-dimensional hollow porous carbon nanofiber as high energy density hybrid supercapacitor, Journal of Energy Storage, 2023, 60, 106713
  2. Milan Babu Poudel, Natarajan Logeshwaran, Ae Rhan Kim, Karthikeyan S.C., Subramanian Vijayapradeep, and Dong Jin Yoo, Integrated core-shell assembly of Ni3S2 nanowires and CoMoP nanosheets as highly efficient bifunctional electrocatalysts for overall water splitting, Journal of Alloys and Compounds, 2023, 960, 170678

3.DOS profile should be presented.

Reply: Thanks very much.

In the revised paper, the DOS is added (shown in Figure 5), and the authors also give some sentences to discuss the results of DOS.

 ‘The projected density of states (PDOS) is also utilized to investigate the atomic interaction and atomic contribution to the molecular orbitals of CoKn clusters. Figure 5 gives the spin polarized projected density of states of CoK4 and CoK5 clusters. Take CoK4 for example. As shown in Figure 4, the molecular orbitals of CoK4 are found in the energy range of -2.0 ~ -2.8 eV (spindle-like spin up molecular orbitals), -3.4 ~ -4.6 eV (petal-shaped spin up molecular orbitals), -4.8 eV (sphere-like spin up molecular orbital), -2.0 ~ -2.4 eV (petal-shaped spin down molecular orbitals), and -4.0 eV (sphere-like spin down molecular orbitals). As shown in Figure 5, in the energy range of -2.0 ~ -2.8 eV, there are K-s, Co-s, and Co-p states (for spin up states), and K-s, Co-d states (for spin down states). That’s to say, the hybrid molecular orbitals coming from the Co-K interaction are found in this energy range, and the K-s states play important role in forming the spindle-like spin up molecular orbitals, the Co-d and K-s atomic states give main contribution in forming the spin down petal-shaped molecular orbitals. In the energy range of -3.0 ~ -4.0 eV, there are mainly the Co-d and K-s states in the spin up states, indicating the spin up petal-shaped molecular orbitals are mainly coming from the Co-d and K-s atomic states. While there are only Co-s states found nearby -4 eV in spin down molecular orbitals, indicating the Co-s states give main contribution in determining the spin down molecular orbitals. There are Co-d and Co-s states found in the energy range of -4.0 ~ -4.6 eV in the spin up molecular orbitals, and there are mainly Co-s states found in the spin up molecular orbitals nearby -4.8 eV. So, the Co-d and Co-s states give main contribution in forming the spin up petal-shaped molecular orbitals in the energy range of -4.0 ~ -4.6 eV, and Co-s states determine the spin up spherical molecular orbitals nearby -4.8 eV. In a word, as for CoK4 cluster, the main components of petal-shaped D molecular orbitals comes from Co-d and K-s states (like D molecular orbitals in the energy range of -2.0 ~ -2.4 eV) or Co-d and Co-s states (like D molecular orbitals in the energy range of -4.0 ~ -4.6 eV). And the main components of sphere-like S molecular orbitals or spindle-like P molecular orbitals are K-s states (like P molecular orbitals in the energy range of -2.0 ~ -2.8 eV) or Co-s states (like S molecular orbitals nearby -4.8 eV). Similar conclusions can also be found in CoK5 cluster. The main components of spindle like P molecular orbitals in the energy range of -2.0 ~ -2.8 eV are K-s states. The main components of spherical S molecular orbitals nearby -4.3 eV and -4.7 eV are Co-s states. The main components of petal shaped D molecular orbitals in the energy range of -3.6 ~ -5.0 eV are Co-d and K-s states (for D orbitals in the energy range of -3.6 ~ -4.2 eV) and Co-d and Co-s states (for D molecular orbitals in the energy range of -4.6 ~ -5.0 eV).’